# New Approaches to Manage Asian Soybean Rust (*Phakopsora pachyrhizi*) Using *Trichoderma* spp. or Their Antifungal Secondary Metabolites

**DOI:** 10.3390/metabo12060507

**Published:** 2022-06-01

**Authors:** Abbas El-Hasan, Frank Walker, Iris Klaiber, Jochen Schöne, Jens Pfannstiel, Ralf T. Voegele

**Affiliations:** 1Department of Phytopathology, Institute of Phytomedicine, Faculty of Agricultural Sciences, University of Hohenheim, Otto-Sander-Str. 5, D-70599 Stuttgart, Germany; 2Central Chemical-Analytical Laboratory, Institute of Phytomedicine, Faculty of Agricultural Sciences, University of Hohenheim, Otto-Sander-Str. 5, D-70599 Stuttgart, Germany; frank.walker@uni-hohenheim.de (F.W.); jochen.schoene@uni-hohenheim.de (J.S.); 3Core Facility Hohenheim, Mass Spectrometry Unit, University of Hohenheim, Ottilie-Zeller-Weg 2, D-70599 Stuttgart, Germany; iris.klaiber@uni-hohenheim.de (I.K.); jens.pfannstiel@uni-hohenheim.de (J.P.)

**Keywords:** biocontrol, *Trichoderma*, antifungal activity, secondary metabolites, 6-pentyl-α-pyrone, harzianic acid, harzianolide, urediospore germination, systemic effect, *Phakopsora pachyrhizi*

## Abstract

Attempts have been made to determine the *in vitro* and *in planta* suppressive potential of particular *Trichoderma* strains (T16 and T23) and their secondary metabolites (SMs) against Asian soybean rust (ASR) incited by *Phakopsora pachyrhizi*. Aside from the previously identified SMs 6-pentyl-α-pyrone (6PAP) and viridiofungin A (VFA), the chemical structures of harzianic acid (HA), iso-harzianic acid (iso-HA), and harzianolide (HZL) were characterized in this study. Our results indicate that exposure of urediospores to 200 ppm 6PAP completely inhibits germination. A slightly higher dosage (250 ppm) of HZL and VFA reduces germination by 53.7% and 44%, respectively. Germ tube elongation seems more sensitive to 6PAP than urediospore germination. On detached leaves, application of conidia of T16 and T23 results in 81.4% and 74.3% protection, respectively. Likewise, 200 ppm 6PAP recorded the highest ASR suppression (98%), followed by HZL (78%) and HA (69%). Treatment of undetached leaves with 6PAP, HA, or HZL reduces ASR severity by 84.2%, 65.8%, and 50.4%, respectively. Disease reduction on the next, untreated trifoliate by T23 (53%), T16 (41%), HZL (42%), and 6PAP (32%) suggests a translocation or systemic activity of the SMs and their producers. To our knowledge, this study provides the first proof for controlling ASR using antifungal SMs of *Trichoderma*. Our findings strongly recommend the integration of these innovative metabolites, particularly 6PAP and/or their producers in ASR management strategies.

## 1. Introduction

Since its discovery in Japan in 1902, Asian soybean rust (ASR) incited by the fungal pathogen *Phakopsora pachyrhizi* Syd. and P. Syd. has been the most challenging foliar disease threating soybean production on a global scale. Under favorable weather conditions this obligate biotrophic fungus can defoliate complete soybean fields within a very short time, leading to up to 90% yield losses [1]. Rust pathogens initiate epidemics through asexual urediospores which germinate under favorable conditions with a sole germ tube followed by subsequent formation of an appressorium [2]. Unlike the vast majority of rust fungi, urediospores of *P. pachyrhizi* utilize an interplay of melanin-independent turgor pressure in appressoria [3] and cell wall degrading enzymes to directly penetrate the epidermis of a host plant rather than entering the leaf tissue through stomata [4,5]. This might shorten the time required for infection and ultimately minimize the exposure of germ tubes to various abiotic and biotic factors. In the following infection stages, infection hyphae grow intercellularly and enter plant cells with haustoria for gaining nutrients. Thereafter, massive infestation of the mesophyll occurs, which culminates in the formation of uredia that emerge through the epidermis, bringing several thousands of wind-dispersed urediospores to the leaf surface, which in turn will be ready to initiate new infections [2,5].

Like other rust fungi, *P. pachyrhizi* completes the majority of its life cycle, apart from spore germination and germ tube extension with subsequent appressoria formation, endophytically [2,5]. In the epiphytic phase, spore germination constitutes the central critical stage required for pathogenesis of *P. pachyrhizi* [6]. It has been frequently reported that eliminating primary infection is favorable over combating subsequent disease development [2,7,8]. In this context, targeting the germination of rust spores or interrupting the elongation of germ tubes seems to be a promising strategy in ASR management. So far, management of ASR has mainly been based on protective and systemic chemical fungicides [9]. Contrary to systemic compounds, protectants do not penetrate the epidermis or translocate within plant tissue [10,11]. Hence, an almost complete coverage of the plant surface and multiple applications are required to protect newly emerging plant tissue during the growing season [12]. However, the repeated use of fungicides with a similar mode of action is increasingly causing resistance of *P. pachyrhizi* to some fungicides [13,14]. This accelerating insensitivity to fungicides, paired with an increasing social awareness about the downsides of synthetic plant protection agents [15] and the urge to reduce environmental risks of associated health hazards [16], has stimulated research towards alternative strategies typically based on biological tools or novel active molecules [17,18,19].

The hyperparasite *Verticillium lecanii* reduces the spread of spores of *Uromyces appendiculatus* on bean by 68% under greenhouse conditions [20]. Saksirirat and Hoppe [21] described the associations of *V. psalliotae* with *P. pachyrhizi* and attributed the primary mode of parasitism to the degradation of urediospores by β-glucanase, chitinase and protease activities. It took more than a decade until Kumar and Jha [22] observed degradation of *P. pachyrhizi* urediospores by *Trichothecium roseum*. The authors explained this lysis with the involvement of similar enzymatic activities. Later, the fungus *Simplicillium lanosoniveum* was found to colonize uredia of *P. pachyrhizi* after their emergence [17], restricting secondary infection pressure and thereby ASR development in the field [23].

Likewise, production of antifungal substances by Biological Control Agents (BCAs) may inhibit the germination of rust spores. Culture filtrates of *Bacillus* sp. and *Streptomyces* sp. showed antifungal and translaminar activity when applied to snap bean inoculated with *U. appendiculatus* in greenhouse experiments [24]. Similarly, *Trichoderma harzianum* RU01 and *Pseudomonas aeruginosa* KMPCH reduced bean rust incidence due to the induction of systemic resistance in common bean under greenhouse conditions [25]. In addition, application of conidial suspensions or sterile culture filtrates of six *T. harzianum* strains resulted in a reduction of uredia formation of *U. appendiculatus* [26]. In a similar observation, propagules of *T. harzianum, B. subtilis*, and *P. fluorescens* reduced urediospore germination of *Puccinia sorghi*, the causal agent of common rust of maize, by 31% to 53% *in vitro* [27]. Recently, El-Sharkawy et al. [18] reported that application of *Trichoderma* spp. alone or in combination with arbuscular mycorrhiza significantly reduced stem rust of wheat incited by *Puccinia graminis* f.sp. *tritici* under greenhouse conditions attributing the antifungal activity of culture filtrates of *T. harzianum* to antibiosis. Furthermore, the strain QST 713 of *B. subtilis*, which is the active ingredient in the registered commercial biofungicide Cease^®^ (BioWorks Inc., Victor, NY, USA), was reported to achieve >90% protection against *P. pachyrhizi* when applied 3 h after inoculation [28]. The authors speculated that the strain might have inhibited spore germination through the production of antibiotics and/or stimulation of host defense responses. Meanwhile, the same strain included in another product (Serenade^®^ ASO, Bayer CropScience, Leverkusen, Germany) used in sequential and alternating applications with the fungicides Pyraclostrobin and Epoxiconazole reduced the area under disease progress curve (AUDPC) up to 69% under low ASR pressure [29].

In earlier studies we have successfully characterized several BCAs—mainly *Trichoderma* spp.—and their secondary metabolites (SMs) active against different plant pathogens [30,31,32,33]. *Trichoderma* spp. have attracted special attention due to their cell wall-degrading capability [34], hyperparasitic ability [35], competition with pathogens for resources [36], induction of defense responses in host plants [37], and excretion of antifungal SMs [38]. Antibiosis as a mode of action by members of the genus *Trichoderma* has originally been described by Weindling and Emerson [39]. Since then, an increasing number of SMs with antifungal properties have been identified [38,40,41,42,43,44]. However, the production of bioactive SMs by *Trichoderma* spp. varies greatly between strains and depends mainly on environmental factors and type and stability of the elicitor used to induce their expression [45]. The large structural diversity of *Trichoderma* metabolites with more than 100 metabolites documented [46] warrants the search for metabolites with a new mode of action. This may be important for the selection of new BCAs, or their novel antifungal SMs for plant protection programs. We have already shown that some antifungal SMs purified from culture filtrates of *Trichoderma* spp. were superior in suppressing different critical developmental stages of *F. graminearum*, i.e., conidia and ascospore germination, perithecia formation, and ascospore discharge [33]. Moreover, strong inhibitory effects of these metabolites against propagules of different phytopathogens, including *F. oxysporum* f.sp. *fabae*, *Alternaria alternata*, *Botrytis cinerea*, *Verticillium dahliae*, *Phytophthora infestans*, and *Sclerotinia sclerotiorum*, have been recorded [32].

To the best of our knowledge, no report is available describing the effect of *Trichoderma* spp. or their bioactive antifungal SMs against ASR. In the present study, SMs purified from *Trichoderma* cultures were tested for their inhibitory potential against urediospore germination and germ tube elongation of *P. pachyrhizi*. Moreover, bioactive SMs, along with live conidia of their producers, were evaluated for their potential to control ASR development both on detached leaves and whole plants. Special attention was given to the question whether reduction of ASR severity by *Trichoderma* strains could be attributed to the excreted antifungal SMs. Furthermore, possible modes of action of *Trichoderma* spp. and their SMs, including antibiosis and systemic/translocation potential, are discussed.

## 2. Results

### 2.1. Identification of the Chemical Structures of SMs from Trichoderma *spp.*

In our earlier studies, the chemical structures of 6-pentyl-α-pyrone (6PAP) and viridiofungin A (VFA) extracted from culture filtrate of *Trichoderma* strain T23 were elucidated [30,32]. Several fractions of strain T16 were collected and their bioactivity against plant pathogens, including *Fusarium* spp., were tested as previously described [32,33]. Based on their biological activities, fractions F116, F416, and F616 from strain T16 were selected and subjected to HPLC/HRMS analysis for chemical structure identification. F116 did not show intense signals in the LC-ESI-MS/MS analysis and has therefore not been characterized further in this study. The LC-ESI-MS chromatogram of the orange, oily fraction F416 exhibited two major peaks with retention times of 7.52 min and 8.40 min (Figure 1a). These two peaks (1 and 2) showed similar mass spectra with molecular ions at *m*/*z* 366.190 ([M + H]^+^) (Figure 1b_1_,b_2_). The latter *m*/*z* value allowed the calculation of the molecular formula of C_19_H_28_NO_6_ for both substances. Literature review revealed that the substance corresponding to this molecular formula might be harzianic acid (HA) [42,47,48,49,50]. In addition, the UV absorption of peak 1 with *λ_max_* 238 and 358 nm (Appendix A) was similar to those found for HA in multiple reports [42,50,51,52]. The identity of HA was confirmed using LC/MS/MS spectra in the positive ionization mode which showed distinct product ions (Figure 1c_1_,c_2_) corresponding to the already predicted mass spectra of HA [47,53]. Moreover, MS/MS data were imported into Compound Discoverer™ 3.3 software (Thermo Scientific, Fremont, CA, USA) to detect chromatographic peaks and the mass of real measured fragments compared with a list of generated theoretical metabolites. A fragment ion search (FISh) score for peak 1 (HA) was used to determine the percentage match between theoretical and experimental structural annotations which measured 70.4%. The latter value is higher than the threshold value of FISh coverage score (≥50) (Appendix A). Finally, comparing the obtained LC/MS/MS spectra with those of the analytical standard harzianic acid (ChemFaces Biochemical Co., Ltd., Wuhan, China) confirmed the elucidated chemical structure (Appendix A). For further characterization of possible isomers, peak 2 with the precursor ion *m*/*z* 366.19040 ([M + H]^+^) was also subjected to MS/MS fragmentation in positive ionization mode. The results showed that MS/MS fragments of this peak was likely identical to that predicted by the first peak (Figure 1c_1_,c_2_), suggesting that this fragmentation pattern was characteristic for an isomer of HA, namely iso-harzianic acid (iso-HA), a naturally occuring biomolecule often accompanying HA [49]. MS/MS fragmentation pattern showed higher ion abundances at *m*/*z* 348, 270, and 252 for iso-HA than HA. The most significant signal at *m*/*z* 348 corresponded to the elimination of water and could be explained with the stereochemistry of iso-HA. Due to the higher abundance of the peak 1, HA was used in the biological trials.

The LC-ESI-MS chromatogram of the brown, oily fraction F616 showed a major peak with retention time of 5.40 min (Figure 2a). Peak 3 showed UV absorption maxima at 220 and 308 nm (Appendix A). LC-ESI-MS/MS analysis of this peak showed a major precursor ion at *m*/*z* 223.13249 (M + H)^+^ corresponding to the molecular formula C_13_H_19_O_3_ (Figure 2c). MS/MS analysis of this precursor ion showed fragmentation pattern (Figure 2d) similar to those of harzianolide found by Cai et al. [44]. MS/MS fragments of peak 3 were computed by Compound Discoverer and auto-annotated with chemical structure, molecular weight, and elemental composition on MS/MS spectra (Appendix A). The calculated FISh coverage score recorded 62.5%, where 30 product ions were successfully matched, and 18 ions remained unmatched. The latter FISh coverage score was above the threshold value (≥50). Based on these data, harzianolide (HZL) could be tentatively identified as the main compound in the fraction F616. However, according to both UV at 310 nm and MS2 spectra, additional structurally analogous molecules almost in acetylated forms with identical UV absorbance and similar fragmentation pathways have been detected (Appendix A). The protonated quasi-molecule ion showed a signal at *m/z* 265.14297 [M + H]^+^ suggesting the molecular formula C_15_H_21_O_4_ (Appendix A).

The mass difference to the [M + H]^+^ signal of the tentatively identified harzianolide with the *m*/*z* 223.13249 (Appendix A) is *m*/*z* 42.01048, which corresponded to the molecular formula C_2_H_2_O. Therefore, it was reasonable to assume that this compound represents an acetylated form of spectrum (a), the harzianolide (protonated molecular formula C_13_H_19_O_3_). The obtained in-source fragmentation at *m*/*z* 223.13235 and 205.12199 showed matching neutral losses of 42 and 60, respectively, confirming the postulated acetylation. Both fragmentation spectra were almost identical (Appendix A) and thus confirmed the closely related chemical structures. Furthermore, structurally similar compounds (c–e) have been detected, all with almost identical fragmentation spectra in the range of *m*/*z* 50–223, the spectrum of HZL (Appendix A). Compound (c) with the [M + H]^+^ signal of *m*/*z* 445.25781 showed a mass difference to HZL of *m/z* 222.1253. The resulting MS/MS spectrum showed first the neutral loss of H_2_O [M + H − 18]^+^ and then further loss of *m*/*z* 222 corresponding to the molecular formula C_13_H_18_O_3_. Based on these apparent similarities, we concluded a dimeric structure of these compounds. Starting from compound (c), two other molecules have been observed. The protonated signal with *m*/*z* 487.26865 of the molecule (d) has a difference of *m*/*z* 42.0108 from the compound (c), indicating acetylation again. Finally, we found one more component of this similar molecules with *m*/*z* 529.2794 (e). The molecular weight showed again the difference of 42.0108, suggesting a second acetylation of the compound (c). In addition, UV spectra of the described molecules were almost identical. Based on these data, we assumed structurally very similar compounds, starting from HZL (Appendix A). Elucidation of the final chemical structures of these similar molecules, which will require much more complex analysis, is beyond the scope of the current study. Being the dominant substance in F616, HZL is considered in the subsequent biological studies.

### 2.2. In Vitro Efficacy of SMs on Urediospore Germination and Germ Tube Development

While the germination rate in 3% acetone (≥95%) did not significantly differ from the water control, urediospores reacted differentially to individual SMs (Figure 3). Germination and germ tube elongation were affected by the different SMs in a concentration-dependent manner. The lowest lethal dose for urediospore germination was recorded for 6PAP (LD_50_ = 58 ppm), followed by HZL (LD_50_ = 222 ppm) isolated from cultures of T23 and T16, respectively (Table 1).

While urediospore germination was almost completely suppressed using 200 ppm 6PAP, a slightly higher dosage (250 ppm) of HZL or VFA reduced germination by 53.7% and 44%, respectively (Figure 3a). Increasing dosages of 6PAP up to 100 ppm resulted in an exponential decrease in the germination of urediospores. Thereafter, this effect steadily increased with rising dosages until it reached a maximum at 200 ppm. Urediospores exposed to dosages of up to 200 ppm HZL, or VFA performed similarly. However, urediospores reacted more sensitive to HZL at higher doses, and 300 ppm of HZL or VFA suppressed urediospore germination by 66.4% or 47%, respectively. HA at concentrations ≥200 ppm was more effective in suppressing urediospore germination than F116. However, both metabolites performed similarly at lower doses and were considered to be less potent inhibitors of urediospore germination.

The mean germ tube length after 6 h of incubation at 25 °C in water or 3% acetone were not significantly different. This indicates no inhibitory effect of acetone on germ tube growth of *P. pachyrhizi* at the concentration used. Similar to the effect on spore germination, germ tube elongation was massively suppressed in the presence of lower concentrations of 6PAP (≤100 ppm, Figure 3b). Similar doses of HZL did not significantly affect germ tube development. However, at higher concentrations of HA or HZL (>100 ppm), germ tube elongation steadily decreased. These actions of both metabolites, however, did not differ between each other. While germ tube growth was more sensitive to 6PAP (LD_50_ = 33 ppm) than urediospore germination (LD_50_ = 58 ppm), HZL and VFA suppressed hyphal growth only at dosages much higher than their LD_50_s in germination (Table 1).

### 2.3. Diphasic Liquid–Solid Fermentation of Trichoderma Strains

For production of *Trichoderma* inoculum, rye and wheat kernels in both full and coarsely ground forms were used as substrates in semi solid-state fermentation. Talcum was used to dry and stabilize the formulation and improve the shelf life of the products. Results indicate that the formulations obtained from a sieving process after 14 d of drying at gradually increasing temperatures up to 38 °C yield a homogenous mixture of talcum and *Trichoderma* spores with 8.7% moisture content. Generally, crushed kernels induced higher spore numbers than intact kernels (Table 2).

Among the different substrates tested, crushed wheat kernels were significantly superior in supporting sporulation of both *Trichoderma* strains T16 and T23 which produced mean propagule counts of 8.1 × 10^8^ and 2.5 × 10^7^ cfu g^−1^, respectively. T16 produced substantially higher spore counts than T23 on all substrates.

### 2.4. Effects of Trichoderma spp. or Their SMs on P. pachyrhizi on Detached Soybean Leaves

To evaluate the influence of *Trichoderma* strains or their SMs on ASR development on detached leaves, conidial suspensions or 200 ppm of SM were applied to the right segments of intact leaves 24 h prior to inoculation with *P. pachyrhizi*. Pooled results obtained from two independent experiments revealed that, under high relative humidity, rust pustules developed well within 12 d on leaves treated with water. No increase or decrease in uredia covered areas (UCAs) was detected in the case of 2% acetone application (Appendix A). When comparing *Trichoderma* strains, application of conidial suspensions of T16 and T23 on segments of soybean leaves reduced the development of rust pustules by 81.4% and 74.3%, respectively (Figure 4). While both strains performed equally well on treated segments, performance of T16 on untreated segments was markedly greater (>1.7 fold), indicating a systemic effect. All SMs tested reduced UCA on treated portions of the leaves significantly, although to different extents (Appendix A). The highest disease suppression was recorded on leaves treated with 200 ppm of 6PAP, generating a more than 2.4 fold better protection than that produced by VFA excreted by the same strain (T23). Among the SMs produced by T16, HZL was the best performing, followed by HA and F116 generating UCA inhibition of 79.7%, 68.9%, and 51.8%, respectively (Figure 4). All metabolites and strains tested also restricted ASR development on untreated parts of the leaves, with inhibition percentages ranging from 7.1% in the case of F116 to 78.4% in the case of 6PAP (Figure 4). This suggests a translocation or systemic activity of the SMs and their producers.

While HA and 6PAP showed high translocation within the leaves, translocation of HZL and F116 was less pronounced. Although ASR suppression caused by HZL on treated segments, was about two times greater than that of VFA, both metabolites differed only insignificantly in terms of their actions on untreated segments (Figure 4).

### 2.5. Validating the Efficiency of Trichoderma and Its Metabolites against P. pachyrhizi under Greenhouse Conditions

To approve the findings obtained from spore germination and detached leaf assays, individual soybean trifoliates were separately sprayed with the best performing secondary metabolites (6PAP, HA, and HZL) or conidial suspensions of *Trichoderma* spp. (T16, T23) 3 h before the entire plant was inoculated with *P. pachyrhizi* under greenhouse conditions. In the control group, where treated plants were not inoculated with urediospores, no harmful phytotoxic effects could be detected (Figure 5a). By contrast, trifoliates treated with the DMI fungicide Osiris^®^ showed some necrotic lesions on the distal parts of soybean leaves (Figure 5b).

Disease severity (represented as % UCAs) in the untreated but inoculated control reached 75%, 14 dpi. Under this heavy infection pressure, all treatments significantly (*p* < 0.05) reduced ASR infection on both treated “first” and untreated “second” trifoliates (Table 3). Osiris^®^ reached 100% and 84.2% control of ASR on treated and untreated trifoliates, respectively. A similar level of protection was achieved on 6PAP-treated leaves. 6PAP was the most effective among the treatments, followed by HA and HZL, which suppressed disease severity by 65.8% and 50.4%, respectively.

All strains and metabolites tested were generally less effective on untreated trifoliates generating inhibition between 18.7% in case of HA and 41.7% in case of HZL. Interestingly, reduction of UCAs on trifoliates treated with conidial suspensions of *Trichoderma* spp. were markedly lower than those produced on the next higher untreated leaves.

## 3. Discussion

Antibiosis is considered one of the most effective biocontrol mechanisms utilized by BCAs, including *Trichoderma* spp. [38,39,40]. In our previous studies, the major antifungal SMs produced by *Trichoderma* strain T23, along with their chemical structures’ elucidation, antifungal activity, and involvement in the antagonist–plant–pathogen interactions, were discussed [30,31,33]. Despite the fact that the strain T16 does not excrete any of the SMs characterized in T23 culture, other compounds with high antifungal activity, including HA, iso-HA, and HZL, were tentatively identified in this study. However, these metabolites have been already detected in cultures of *T. harzianum* [43,44,45,47,54]. In addition, dimeric compounds, structurally very similar to HZL in acetylated forms, with identical UV absorbance and similar fragmentation spectra, have been detected in fraction F616. The final chemical structures of these analogous molecules, along with the bioactive fraction (F116), have not been yet characterized and will be a subject for further analysis.

Due to the lack of reports describing the effect of *Trichoderma* spp. or their antifungal SMs against ASR, efforts have been made to investigate the suppressive potential of selected *Trichoderma* strains and their purified SMs against this pathogen both *in vitro* and *in vivo*. Given the integral role of urediospores to generate ASR epidemics, we initially examined the processes of urediospore germination of *P. pachyrhizi* and germ tube elongation in the presence of selected SMs purified from *Trichoderma* spp. Our results indicate that exposure of urediospores to SMs substantially suppress germination and germ tube growth (Figure 3). The most potent inhibitor in both cases was found to be 6PAP, followed by HZL. Urediospore germination and germ tube elongation were strongly impaired by 6PAP in a dose-dependent manner. While the high sensitivity of urediospores to 6PAP (at 200 ppm) is comparable with that of macroconidia and ascospores of *F. graminearum* [33], suppression of microconidia of *F. moniliforme* required a higher dosage (250 ppm) [30]. By contrast, a concentration of 220 ppm of 6PAP reduced spore germination of *F. oxysporum* f.sp. *lycopersici* by only 34% [55]. This suggests a remarkable diversity of responses to a germination inhibitor among spore types and fungal species. Urediospores of *P. pachyrhizi* and macroconidia of *F. graminearum* showed a relatively similar sensitivities to VFA, causing inhibition of germination at 200 ppm of 44% and 56.8%, respectively [33]. Germ tube elongation from urediospores, on the other hand, was two times greater compared to germ tube expansion from macroconidia. Pronounced inhibitory effects of VFA on the germination of conidia of *F. oxysporum* f.sp. *fabae*, *A. alternata*, and *B. cinerea* have been observed [32]. In addition, a substantially greater antifungal effect of VFA against conidia of *V. dahliae*, sporangia of *P. infestans*, and sclerotia of *S. sclerotiorum* was observed [32]. This finding clearly provides evidence for the wide inhibitory effect of this compound. Urediospore germination was suppressed by 66% in the presence of 300 ppm HZL (formerly F616 [32]). While the LD_50_ of this metabolite (222 ppm) is higher than that of 6PAP (58 ppm), this value is significantly lower than those of VFA and HZL (Table 1). HZL was found to generate pronounced inhibitory effects against *Cladosporium* sp. and *F. moniliforme* [32]. Similarly, this biomolecule inhibited mycelial growth of *Gaeumannomyces graminis* var. *tritici*, *Rhizoctonia solani*, and *Pythium ultimum* to different extents [56]. To our knowledge, the activity of HZL as a potent inhibitor of spore germination is reported for the first time.

Urediospores were less sensitive to HA (formerly F416 [32], LD_50_ = 713 ppm). Contrary to its effect on urediospore germination, similar dosages of HA reduced conidia germination and germ tube growth of *F. graminearum* by 64% and 24%, respectively [33]. These observations might illustrate different levels of spore sensitivity to HA among biotrophic and pertotrophic plant pathogens. However, similar reports on the effect of HA against *Pythium irregulare, S. sclerotiorum,* and *R. solani* have already been described [43].

With the exception of 6PAP, elongation of germ tubes seems to be more sensitive to all other SMs than urediospore germination. This phenomenon could be attributed to the fact that spore germination does physiologically differ from germ tube growth due to the ability of spores to mobilize stored reserves (i.e., proteins, lipids, etc.) for further implementation in the various metabolic pathways [57].

Encouraged by the promising activities *in vitro*, we tested whether SMs and their producers can reduce ASR disease. To this end, a sufficient quantity of *Trichoderma* propagules with high numbers of colony forming units and good quality are produced using crushed wheat kernels as a substrate (Table 2). Since aeriation is considered a key factor for fungal sporulation [58], the permeable PE-bags used in our study ensured high biomass production. The obtained spore counts (≥2.5 × 10^7^ cfu g^−1^) are comparable with that produced by *T. harzianum* cultivated on similar substrates [59,60].

A detached leaf assay was designed using soybean leaves treated only on the right side of the midrib with SMs or conidia of *Trichoderma* strains, 24 h prior to inoculation with *P. pachyrhizi*. Disease severity was quantified using ImageJ software. This powerful tool does provide much more accurate UCA calculation compared with the visually-aided traditional disease rating scales described in several reports [61,62,63]. The presence of antagonists alone initiated neither disease symptoms nor phytotoxic damage. Interestingly, both strains (T16 and T23) substantially suppressed ASR development on treated and untreated leaf areas (Figure 4). Application of conidial suspensions of T16 or T23 protected leaves by 81.4% and 74.3%, respectively. These values represent a much higher protection compared with that (≤58.2%) reported for commercially available *Trichoderma* strains (T39 from Trichodex^®^, T50 from Trichosan^®^, and *T. harzianum* from UniSafe^®^) applied in a similar way on bean leaf disks against *U. appendiculatus* [26]. Burmeister and Hau [26] speculated that bean rust severity in the presence of the protective treatment with undiluted culture filtrates of UniSafe^®^ 24 h before inoculation with *U. appendiculatus* differed insignificantly from those developed after simply applying fungal conidia of the same strain. In a similar scenario, the HZL-producer T16 in our study suppressed ASR on treated leaves to an equal extent as pure HZL itself (Figure 4). Similarly, the strong performance of T23 (>74%) in arresting ASR might be entirely attributed to the secretion of its powerful metabolites 6PAP and VFA which restrict UCA by 98% and 41%, respectively. These results strongly suggest that suppression of ASR by T16 and T23 is likely due to antibiosis, which is in accordance with the finding of Govindasamy and Balasubramanian [64] who found that a pre-treatment of detached groundnut leaves with conidia and concentrated germ fluid of *T. harzianum* significantly suppressed *Puccinia arachidis*. Nevertheless, this does not rule out the possibility that other modes of action could also be involved. The efficiency of T16 and T23 on untreated portions of the leaf might be attributed to the rapid colonization by *Trichoderma* hyphae, bringing the antagonist in direct contact with the pathogen, or to a systemic effect which might have been induced in the adjacent untreated tissue. Such activity of *Trichoderma* spp. against ASR is in agreement with the results obtained by Abeysinghe [25], who reported a partial control (≤50%) of *U. appendiculatus* on common bean using *T. harzianum* RU01, attributing this effect to the induction of systemic resistance.

While 6PAP and HA are considered superior ASR inhibitors on treated and untreated leaves, HZL showed greater inhibition on treated than untreated leaves (Figure 4). This indicates a stronger systemic potential of 6PAP and HA compared with the other metabolites. Similarly, application of 6PAP showed enhanced systemic defense reactions in maize against *F. moniliforme* [31]. However, this should not exclude alternative scenarios for the feasibility of compounds’ translocation from the treated into the adjacent untreated leaf tissue or saturating the phyllosphere with the vaporized metabolites (e.g., 6PAP, HA) inside the tightly closed humid chambers. The latter SMs earlier showed pronounced volatile activity [32]. These three possible mechanisms are currently under further investigation.

Unlike the outcome on detached leaves, both strains (T16, T23) showed greater protection of untreated (41.4%, 53.4%) than treated (26.3%, 38.2%) trifoliates using whole plants in our experiments (Table 3). These results again illustrate potential systemic effects of *Trichoderma* strains. Similarly, application of *Trichoderma* spp. alone or in combination with arbuscular mycorrhiza reduced *Puccinia graminis* f.sp. *tritici* under greenhouse conditions [18]. Moreover, the sequential applications of *B. subtilis* and fungicides reduced the AUDPC of *P. pachyrhizi* by 69% under a low ASR incidence [29]. In our study, a higher activity of both strains on detached compared to that on attached leaves has been observed (Table 3 vs. Figure 4). Reasons for this finding could be either the shorter incubation period *in vivo*, thereby the conidia failed to establish itself and excrete their arsenal of SMs, or the conidia were partially washed out from the attached leaves during the subsequent short-term (3 h) application of the aqueous urediospore suspension.

Application of 200 ppm 6PAP on attached leaves resulted in the highest protection (>84%) against *P. pachyrhizi*. A similar UCA inhibition was recorded on the upper untreated trifoliate next to the leaves sprayed with the systemic fungicide Osiris^®^ (Table 3). Furthermore, a phytotoxic injury in the form of necrotic spots has been observed on leaves treated with this triazole-based fungicide. It has been reported that DMI fungicides including triazoles can cause phytotoxic damage on some soybean varieties, particularly under water stress or elevated temperatures [65,66]. However, Osiris^®^ with protective and curative activities is registered for controlling cereal rusts. Its phytotoxicity on some soybean varieties might be the reason why it is not recommended for ASR management.

6PAP performance on untreated attached trifoliates (Table 3) was less pronounced compared with that on untreated detached leaves (Figure 4). This could be attributed to (*i*) the lack of time essential for triggering the likely metabolite-mediated defense responses before challenge with the pathogen, (*ii*) the readily vaporization of 6PAP when applied under the absence of an equilibrium vapor pressure, (*iii*) the 6PAP-distribution in the various plant parts (e.g., stem), which in turn results in lower concentrations in the untreated trifoliate, or (*iv*) the minimal compound uptake by the leaves and hence decelerating its translocation to the next upper trifoliate. In a comparable report, culture filtrates from *Bacillus* sp. CA5 and *Streptomyces* sp. CS35 were found to have apparent translaminar and leaflet-to-leaflet translocation but no trifoliate-to-trifoliate movement when applied on snap beans inoculated with *U. appendiculatus* under controlled conditions [24].

Among the SMs tested, the auxin-like HZL exhibited the greatest suppression of ASR on the next upper untreated trifoliate. This metabolite seems to be easily delivered within 3 h from the first (treated) to the second (untreated) trifoliate, generating a slightly lower protection (41.7%) compared with the treated trifoliate (50.4%) (Table 3). This might indicate a rapid translocation between trifoliates which ultimately might speculate that apoplastic, rather than symplastic, movement has been utilized by HZL. In this regard, it is well documented that apoplast is the transportation route for several phytohormones including auxins and cytokines [67,68]. In addition, it has been reported that the apoplast is the place where MAMPs are secreted by pathogens [69]. This suggests that *Trichoderma* metabolites could interfere with a much more complex molecular cross-talk with *P. pachyrhizi.* Cai et al. [44] showed that a subsequent challenge of HZL-pretreated tomato plants with *S. sclerotiorum* resulted in a higher expression of defense-related genes involved in salicylic acid and jasmonate/ethylene signaling pathways.

While HA performed equally on treated and untreated detached leaf portions (Figure 4), its influence on the treated attached trifoliates was 3.6-fold greater than those on untreated trifoliates (Table 3). HA has been reported as a siderophore, having a high affinity for Fe^3+^ capable to scavenge iron from the environment, and hence depriving the pathogens of this prerequisite micronutrient [64]. This novel effect might explain the higher efficiency on the closely adjacent segments of detached leaves than that on the attached leaves, which are spatially distant from the treatment.

Finally, it has been frequently reported that rainfall or overhead sprinkling can modify fungicide deposits on plants by dilution, dissemination, or removal [70,71]. Surprisingly, washing off the leaves with aqueous urediospore solution only 3 h after leaves had been sprayed with SMs or conidial suspensions of *Trichoderma* spp. resulted in a considerable protection against *P. pachyrhizi*. This discovery indicates an excellent rain fastness, which in turn might strongly enhance the performance and durability of the biocontrol.

## 4. Materials and Methods

### 4.1. Fungal Material

*Phakopsora pachyrhizi* race Thai 1 [7], *Trichoderma harzianum* (T16, GenBank ACNO: MW520837), and *T. asperellum* (T23, GenBank ACNO: MW509067) were obtained from the culture collection of the Institute of Phytomedicine, University of Hohenheim. Urediospores of *P. pachyrhizi* were multiplied by inoculating 14-day-old soybean plants (cv. Thorne, Bayer CropScience, Lyon, France) at 22 °C and a 16 h light/8 h dark regime. Freshly produced urediospores were harvested 14 days post-inoculation (dpi) and stored at −80 °C. *Trichoderma* strains were maintained in sterilized soil cultures at 4 °C. Before being used, strains were sub-cultured on potato dextrose agar (PDA; Carl Roth, Karlsruhe, Germany) and allowed to grow at 25 °C in the dark for 7 d.

### 4.2. Production, Extraction, and Purification of the Secondary Metabolites from Trichoderma Cultures

*Trichoderma* metabolites were produced by inoculating 1 L potato dextrose broth (PDB; Sigma-Aldrich Chemie, Munich, Germany) in 2 L conical flasks with 30 agar plugs (5 mm Ø) taken from actively growing cultures of the respective *Trichoderma* strains in triplicates. After 12 d incubation on a rotary shaker at 150 rpm and 25 °C in the dark, the fungal biomass was removed by filtration through a Whatman filter paper and the culture filtrates were extracted twice with equal volumes of ethyl acetate (EtOAc; Merck KGaA, Darmstadt, Germany). The organic phases were combined, dehydrated with Na_2_SO_4_, and evaporated under reduced pressure (150–170 mbar) in a rotary evaporator at 40 °C. The residues were then re-dissolved in a mixture of acetonitrile (ACN): H_2_O [1:1 (v:v)] and centrifuged at 12,000 rpm for 5 min. Supernatants were eventually submitted to Varian^®^ preparative HPLC. The purified compounds were separately collected, pooled, extracted with EtOAc, and eventually evaporated until the substances recrystalized. For biological assays, the purified metabolites were re-dissolved in acetone (stock solution 10 mg mL^−1^) according to El-Hasan et al. [33].

### 4.3. Chemical Structure Elucidation of the Secondary Metabolites

The chemical structures of the metabolites (100 mg L^−1^ in acetonitrile (ACN): H_2_O (1:1)) were elucidated by HPLC/MS and Q-Exactive^TM^ Plus Hybrid Quadrupol-Orbitrap™ mass spectrometry (Thermo Fisher Scientific GmbH, Bremen, Germany) equipped with a heated electrospray ionization source. Chromatographic separation was performed at a temperature of 40 °C using an Agilent Zorbax Eclipse Plus column (2.1 mm × 50 mm × 1.8 μm; Agilent, Waldbronn, Germany). The mobile phase consisted of 0.1% formic acid in water (A) and 0.1% formic acid in ACN (B). The pressure limit ranged from 0–1200 bar and the linear gradient was set as follows: 0–1 min, 90% A and 10% B; 1–13 min, 5% A and 95% B; holding for 2 min and recondition at 16.00 min, 50% A and 50% B. The flow rate was 0.4 mL/min with an injection volume of 5 μL. The scan mass range was set at *m*/*z* 140–1200. The parameter for ESI (+) measurements were set as follows: a full scan and fragment spectral resolution were 70,000 FWHM and 17,500 FWHM, respectively; the capillary temperature was 360 °C; auxiliary gas heater temperature was 350 °C; spray voltage was 4.2 kV; sheath gas flow rate was 60 Arb; auxiliary gas flow rate was 20 Arb; S-lens RF level was set at 50. The acquisition mode of stepped normalized collision energy (NCE) was used with settings of 20, 60, and 110 eV. Confirmation of the expected compounds was done using the fish score node of Compound Discoverer (Version 3.2, Thermo Scientific, Fremont, CA, USA).

### 4.4. Preparation of Urediospore Suspensions

Germination rates of urediospores were determined and those reaching at least 80% germination were selected for further use. Urediospore suspensions of *P. pachyrhizi* were prepared according to the method of Hirschburger et al. [72] with minor modifications: 200 mg urediospores were re-hydrated in 100 mL sterile distilled water containing 200 mg milk powder (Carl Roth) and 10 µL Tween^20^ (Th. Geyer, Renningen, Germany). This resulted in a final concentration of 2.3 × 10^4^ urediospore mL^−1^ determined using a hemocytometer (Brand™ Fuchs-Rosenthal; Fisher Scientific, Schwerte, Germany). The urediospore suspensions were subsequently incubated at 29 °C on a rotary shaker with 250 rpm under dark conditions for 30 min.

### 4.5. Impact of Metabolites on Urediospore Germination and Germ Tube Elongation In Vitro

To determine the influence of *Trichoderma* metabolites (6PAP, VFA, HA, HZL, and F116) on urediospore germination and germ tube elongation of *P. pachyrhizi in vitro*, urediospore suspensions were mixed with metabolites to get a series of final concentrations ranging from 50 to 300 mg L^−1^ in triplicate. Control treatments included 3% acetone and sterile distilled water. Mixtures were sprayed onto polyethylene (PE) sheets (Rische and Herfurth, Hamburg, Germany) and placed on microscope slides, which were immediately transferred to humidity chambers. These were sealed and kept in the dark at 25 °C for 6 h. Germination was stopped with 2% aniline blue in lactophenol (Carl Roth). A urediospore was counted as germinated when the length of the germ tube (LGT) was not less than the width of the urediospore. Percentages of germinated urediospores and LGTs were analyzed using an Axioskop microscope equipped with an Axiocam camera and AxioVision SE64 Rel. 4.8 software (Carl Zeiss AG, Oberkochen, Germany). LD_50_ values were computed using probit analysis [73].

### 4.6. Starter Culture and Diphasic Liquid–Solid Fermentation of Trichoderma Strains

Starter cultures for both *Trichoderma* strains were prepared by individually inoculating 100 mL sterilized PDB in a 300 mL Erlenmeyer flask with 10 agar plugs (5 mm Ø) of a 48 h old *Trichoderma* culture (Figure 6B). Cultures were subsequently incubated on a rotary shaker at 150 rpm and 22 °C in the dark for 72 h to be eventually used as inoculum for solid-state fermentation (SSF). In SSF, four types of substrates (rye and wheat kernels in both full and coarsely ground forms) were evaluated for spore production. Kernels were coarsely ground using a cereal mill. Briefly, 250 g of each substrate were transferred after two consecutive autoclaving steps at 121 °C for 30 min into a 3.5 L autoclavable Microsac^®^ gas-permeable plastic bag (Sac O2, Deinze, Belgium). Substrates were individually inoculated with 50 mL of starter culture and sealed with an impulse heat sealer. Thereafter, cultures were periodically homogenized through manual massage and kept at 27 °C in the dark for 10 d (Figure 6C).

After mixing with an equal amount of talcum (Sigma-Aldrich Chemie), cultures were transferred to loosely closed boxes and air dried at gradually increasing temperatures at 1 °C per 24 h for 14 d (Figure 6D,E). Moisture content was determined according to the formula of Bonner [74]:*Mc**(%)* = *(wW* − *dW)* × *100/wW*
where *Mc*: % of moisture content; *wW*: wet weight (g); *dW*: dry weight (g).

Finally, chlamydospores and conidia were collected by passing through a set of sieves (140–180 µm Ø; Retsch, Haan, Germany) placed on a shaker at 100 rpm (Figure 6F,G). Before storing at 23 °C, spore concentration per g product was determined using a hemocytometer.

### 4.7. Effect of Trichoderma or Its SMs on P. pachyrhizi on Detached Leaves

A detached leaf assay was established to evaluate the effect of *Trichoderma* strains or their SMs on ASR development *in vivo*. Fully developed symptomless trifoliates were cut at the base of the petiole, wrapped with a piece of cotton (200 mg) soaked with 2 mL of 1% Wuxal^®^ Super (8% N, 8% P_2_O_5_, 6% K_2_O and micronutrients; Aglukon, Düsseldorf, Germany), and individually placed abaxial side down on moist towels in an airtight box. Subsequently, 300 µL of each metabolite solution (200 ppm) or an equal volume of conidial suspension (2.5 × 10^7^ conidia mL^−1^), were spread on the right side (according to the proximal-distal axis) of the adaxial leaf surface using a 10 mm sterile paintbrush (Appendix A). Sterile water and 2% acetone served as controls. Leaves were incubated at 23 °C for 24 h in the dark. Thereafter, each leaf received 500 µL of an urediospore suspension and was kept in a growth chamber at 25/20 °C under a 12/12 h light/dark regime for 12 d. Leaves were then photographed and the uredia covered areas (UCAs) were quantified using ImageJ software v. 1.41 (National Institutes of Health, Maryland, USA).

### 4.8. Evaluating the Performance of Trichoderma or Its SMs against P. pachyrhizi under Greenhouse Conditions

Plants were grown in pots (15 cm Ø) containing classic soil (CLT, Einheitserde Werkverband, Sinntal-Altengronau, Germany) until the third trifoliate was completely developed (BBCH 14). To avoid cross-contamination, the whole plant was covered during spraying, with only the oldest trifoliate accessible for treatment with 2 mL of either a 200 mg L^−1^ solution of a metabolite (6PAP, HA, HZL) or a conidial suspension (2.5 × 10^7^ conidia mL^−1^) of T16 or T23. The fungicide Osiris^®^ (BASF SE, Ludwigshafen, Germany) was used as a positive control, whereas spraying with 2% acetone or sterile distilled water served as negative controls. After 3 h, plants were divided into two groups. While plants of the first group were uniformly inoculated with a freshly prepared urediospore suspension of *P. pachyrhizi*, the second group was sprayed with a similar suspension lacking urediospores. Subsequently, plants were incubated in PE-boxes (95 × 60 × 50 cm) at ≥95% relative humidity and 22 °C in the dark for 12 h. Thereafter, they were transferred to normal greenhouse conditions at approx. 50% humidity and a day/night regime of 16 h/8 h at 22 °C for 14 d. UCAs on the oldest trifoliates were quantified using ImageJ. Additionally, possible systemic/translocation effects were assessed by comparing UCAs developed on the second trifoliates (untreated). The experiment was conducted twice with three biological replicates, each consisting of a pot containing two plants.

### 4.9. Statistical Data Analysis

If not stated otherwise, experiments were conducted at least twice in triplicate in a completely randomized design. Statistical analyses were carried out using IBM^®^ SPSS^®^ Statistics software v. 26 (IBM, Armonk, NY, USA). Data were analyzed using one-way analysis of variance (ANOVA). Significant differences were subsequently computed by the post-hoc multiple comparison assay based on Tukey HSD, Duncan, or LSD tests at *p* < 0.05 [75].

## 5. Conclusions

A better understanding of the effect of BCAs and their bioactive SMs on hardly managed biotrophic plant pathogens like rust fungi substantially supports the development of safe and efficient plant production products. In the present study, production, characterization, and bioactivity of antifungal SMs (e.g., 6PAP, VFA, HA, and HZL), along with their producers (T23 and T16) against ASR, are described.

Since spore germination is required for pathogenesis, this process could identify fungus-specific targets for the management of fungal spore-mediated diseases, e.g., ASR. Therefore, here we provide innovative tools for targeting urediospore germination and the further development of *P. pachyrhizi* both at laboratory and whole plant levels. Novel evidence for the activity of SMs purified from *Trichoderma* spp., particularly 6PAP and HZL as potent inhibitors of urediospore germination of *P. pachyrhizi*, are provided. The tested SMs and their living producers proved to have a strong potential to restrain ASR, which was confirmed in both detached leaves and whole plant experiments. Their promising antifungal effects, possibly by combating the pathogen before penetration together with their rapid translocation or activation of systemic defense reactions in soybean, are discussed. These results may not only enlarge our knowledge about the mechanisms involved in biocontrol programs but also put forward future perspectives for integrating *Trichoderma* spp. and/or their SMs with different modes of action as effective biocontrol agents against ASR.

## Figures and Tables

**Figure 1 metabolites-12-00507-f001:**
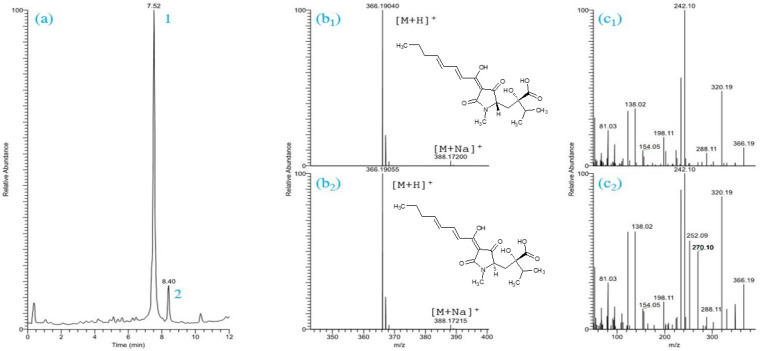
ESI(+)LC/MS total-ion current (TIC) trace of fraction F416, where 1 and 2 are peak No. (**a**), ESI(+)LC/MS mass spectra and chemical structures of harzianic acid (**b_1_**) and iso-harzianic acid (**b_2_**), ESI(+)LC/MS/MS product-ion mass spectra of HA (**c_1_**), and iso-HA (**c_2_**).

**Figure 2 metabolites-12-00507-f002:**
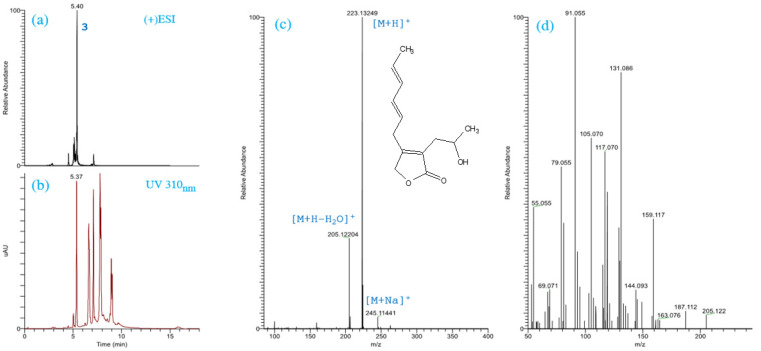
ESI(+)LC/MS (XIC, *m*/*z* 223.13249) trace, where 3 is peak No. (**a**), UV trace at 310 nm (**b**), ESI(+)LC/MS mass spectra and chemical structure of harzianolide (**c**), ESI(+)LC/MS/MS product-ion mass spectra of harzianolide (*m*/*z* 223.13249; [M + H]^+^) (**d**).

**Figure 3 metabolites-12-00507-f003:**
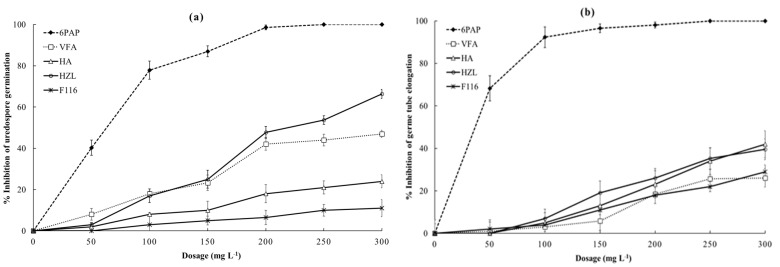
Suppression of urediospore germination (**a**) and germ tube elongation (**b**) of *P. pachyrhizi* in the presence of different concentrations of various SMs purified from *Trichoderma* strains T16 and T23. Urediospores were mixed with SMs to get a series of final concentrations ranging from 50 to 300 mg L^−1^. Mixtures were incubated on microscope slides at 25 °C in the dark for 6 h. Presented data are means of three replicates ± standard error (SE).

**Figure 4 metabolites-12-00507-f004:**
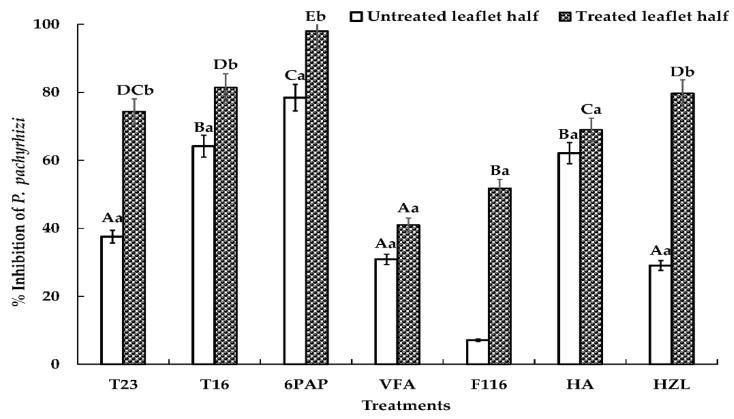
Effects of treatments with conidial suspensions or SMs of *Trichoderma* spp. applied 24 h in advance of *P. pachyrhizi* inoculation on the percentages of uredia covered areas (UCAs) of soybean leaves. Different uppercase letters indicate significant differences among treatments within the same half of the leaf, while different lowercase letters indicate significant differences between treated and untreated halves of the same leaf using LSD test at *p* = 0.05.

**Figure 5 metabolites-12-00507-f005:**
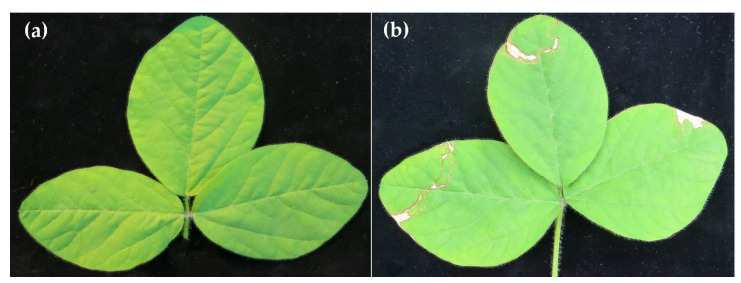
Soybean trifoliates treated with 6PAP (**a**) or Osiris^®^ (**b**) and incubated undetached for 14 d under greenhouse conditions.

**Figure 6 metabolites-12-00507-f006:**
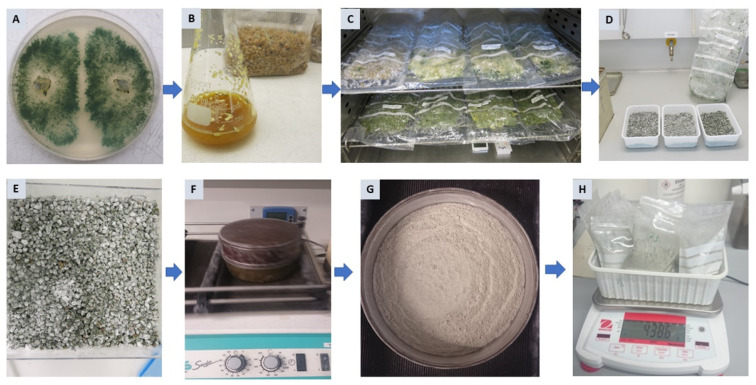
The various stages of production of *Trichoderma* formulations: (**A**) *Trichoderma* mother culture on PDA; (**B**) Starter culture production and inoculation of substrates; (**C**) Incubation of different substrates; (**D**) Drying after mixing with talcum; (**E**) Dried substrates; (**F**) Spores harvest by sieving; (**G**) Product collection; (**H**) Product packaging.

**Table 1 metabolites-12-00507-t001:** Doses generating 50% inhibition (LD_50_) of urediospore germination and length of germ tube (LGT) of *P. pachyrhizi* after 6 h of exposure to secondary metabolites of *Trichoderma* spp. at 25 °C under ≥95% relative humidity (RH) and complete darkness. LD_50_ values are means of two independent assays.

Metabolite	Producing Organism	Strain	LD_50_ (ppm)	
			Germination	LGT
6PAP	*T. asperellum*	T23	58 ^a^	33 ^a^
VFA	*T. asperellum*	T23	299 ^b^	542 ^b^
HA	*T. harzianum*	T16	712 ^c^	346 ^c^
HZL	*T. harzianum*	T16	222 ^d^	394 ^d^
F116	*T. harzianum*	T16	2118 ^e^	609 ^e^

Data were analyzed by one-way ANOVA, followed by post-hoc multiple comparison analysis (Tukey HSD, Duncan, or LSD) at *p* = 0.05. Different letters indicate significant differences.

**Table 2 metabolites-12-00507-t002:** Propagule counts of *Trichoderma* strains T16 and T23 produced on different substrates in semi solid-state fermentation.

Substrate	Strain	CFU (Spore g^−1^)
Whole rye kernels	T16	3.4 × 10^7^ ± 0.57 × 10^7^ _Ba_
	T23	2.6 × 10^6^ ± 2.15 × 10^6^ _Aa_
Crushed rye kernels	T16	1.5 × 10^8^ ± 0.43 × 10^8^ _Bc_
	T23	1.3 × 10^7^ ± 1.31 × 10^7^ _Ab_
Whole wheat kernels	T16	9.2 × 10^7^ ± 0.64 × 10^7^ _Bb_
	T23	1.7 × 10^7^ ± 2.07 × 10^7^ _Ab_
Crushed wheat kernels	T16	8.1 × 10^8^ ± 3.12 × 10^8^ _Bd_
	T23	2.5 × 10^7^ ± 0.86 × 10^7^ _Ac_

Values are means of three replicates ± SE. Data were analyzed by ANOVA followed by Tukey, Duncan, or LSD tests at *p* = 0.05. Different capital letters indicate significant differences between strains on the same substrate, different lowercase letters indicate significant differences of one strain between different substrates.

**Table 3 metabolites-12-00507-t003:** Influence of *Trichoderma* strains and their secondary metabolites on ASR development on treated and next upper, untreated trifoliates.

Treatment	Treated Trifoliates		Untreated Trifoliates	
	% UCA	% Inhibition	% UCA	% Inhibition
T16	27.70 ± 2.21	26.31 ^a^	16.37 ± 2.03	41.41 ^c^
T23	23.25 ± 4.72	38.16 ^b^	13.03 ± 3.87	53.36 ^d^
6PAP	11.90 ± 1.48	84.18 ^e^	49.70 ± 2.31	31.74 ^b^
HA	25.70 ± 2.86	65.82 ^d^	59.20 ± 0.96	18.68 ^a^
HZL	18.95 ± 3.07	50.39 ^c^	16.28 ± 5.08	41.73 ^c^
Osiris^®^	00.00 ± 0.00	100.00 ^f^	11.50 ± 0.76	84.20 ^e^

Values are means of three replicates ± SE. Data were analyzed by one-way ANOVA, followed by post-hoc multiple comparisons analysis (Tukey HSD, Duncan, or LSD at *p* = 0.05). Different letters indicate significant differences among treatments.

## Data Availability

All data generated during the study are included in the article or Appendix A provided.

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
