# Peer review of "New Approaches to Manage Asian Soybean Rust (Phakopsora pachyrhizi) Using Trichoderma spp. or Their Antifungal Secondary Metabolites"

_metabolites, 2022, doi:10.3390/metabo12060507_

Round 1

Reviewer 1 Report

The reported study shows a significant inhibition of spore germination and rust disease development upon treatment with extracts and some isolated compounds from Trichoderma fungi. The main idea in such studies is to provide rationale for biofungicidal applications. Many of the rust-causing species are easily combated using synthetic fungicides but for organic farming, an alternative is highly demanded. There have been plenty of similar studies involving various beneficial and pathogenic fungi, including Trichoderma harzianum as well as many pathogens. However, the originality of this study is that the combination of Phakospora spore gemination and outbreak of disease has not been tested with trichoderma products. As a result, the Authors were able to demonstrate that the specific metabolite as well as the crude preparations can excellently prevent urediospore germination and thus a development of disease symptoms, both in vitro and on detached leaf-assay.

The manuscript is well-written with a comprehensive and smooth Introduction, well-illustrated results section and satisfactorily deep discussion. Therefore, publication of this submission may be considered by the Metabolites journal upon some necessary improvements and responding to some uncertainties.

So, I recommend minor revision, and these are my specific remarks:

1) page 3, line 100 – the abbreviation SM (for "secondary metabolites") appear here for the first time (except abstract), so it should be explained

2) line 142 – what is "letter m/z"?

3) line 146. I wouldn't agree that the identity of HA was "finally confirmed" using LC-MS and spectra alignment with libraries. If not confirmed by standards or other techniques allowing full structure elucidation this identification still remains "tentative". It does not compromise the study but should be borne in mind and properly discussed, as the spectra of isoHA and HA are similar and there is no proof in the study which one is which (even if we can assume it from the peaks proportions)

4) page 4, line 154 – what do you mean by almost accompanied?

5) line 156 – I was a little confused by the information that HA and iso-HA "were referred as HA"? what does it mean exactly and why so?

6) page 5 line 169-170 – again, what do you mean by almost in acetylated forms? also, what do you mean by "structurally analogous oligomers"? A compound with m/z 445, so a putative dimer? Or else? Please explain in the responses and develop this idea in the discussion.

7) line 183 -instead of "antifungal effect" use "this effect" (or "germination-inhibiting effect", to be more specific).

8) line 185 and several other sentences throughout – the phrase "seemed to" is used in an inappropriate context. here, you have specific experimental results, so it shouldn't sound like a speculation or apparent response, as the verb "seem to" suggests. Please rephrase here and in other sections.

9) lines188-189 – also here, the phrasing isn't proper. metabolites did not "behave" here, and actually, shouldn't be "judged". Please rephrase.

10) Table 1. I would like to understand how the LD50 were calculated from dose response curves that didn't even reach 50% and their course wasn't really showing any symptoms of being typically sigmoidal, so it is just an extrapolation (and not really valid). Did you perform any statistical validation of these calculations? ANOVA and post hocs multicomparisons do not really apply to LD50 calculations (rather to each individual means/medians). By the way, which approach was indeed used to indicate differences in LD50s? Please, explain the statistics thoroughly.

11) page 7, line 252 – you wrote about tissue translocation of certain metabolites. However, I couldn't find these results in your study. Please explain how you estimated the translocation/transport of the metabolites to be able to claim these differences?

12)  the structures of all individual metabolites would be much useful in the supplementary file, please draw them and paste into the supplementary file.

Author Response

Dear Reviewer,

Thank you very much for reviewing our manuscript and the valuable comments provided.

Please find our point-by-point response in the attached file. 

Sincerely

The Authors

Reviewer 2 Report

The authors showed the effects of various Trichoderma strains and their secondary metabolites against ASR. I think the manuscript is well written, especially the introduction part.

I have only some questions:

  1. Why do they apply three different kinds of statistical tests?
  2. In many cases, Fig 3 and all tables, the repletion of the experiments are not clear and SD or SE are not indicated.
  3. How were applied metabolites on detached leaves, under darkness or under the normal photoperiod?
  4. When were the experiments started in the case of intact plants, in the early or late dark period?
  5. It remained uninvestigated the self-effects of metabolites on intact plants. Changes in the upper leaves suggest systemic effects. Why did not investigate some physiological parameters of plants e.g. growing rate, weight, or chlorophyll content?

Author Response

(The authors gave the same response as above.)

Reviewer 3 Report

The manuscript metabolites-1707467 attempts to prove the antifungal potential of Trichoderma strains and their secondary metabolites against Asian soybean rust incited by Phakopsora pachyrhizi.

The work is novel, well organized and structured, the number of tables and figures is adequate, and the conclusions are presented in a clear, concise and logical way.

I have several comments on the manuscript, as follows:

  1. Identification of main secondary metabolites from Trichoderma sp.: LC-MS analysis allows the tentative identification of compounds. Identification should be done by comparison with authentic standard or by means of NMR analysis (as made by the authors in previously published papers, e.g. Abbas El-Hasan & Frank Walker & Jochen Schöne & Heinrich Buchenauer. Detection of viridiofungin A and other antifungal metabolites excreted by Trichoderma harzianum active against different plant pathogens. Eur J Plant Pathol (2009) 124:457–470, DOI 10.1007/s10658-009-9433-3).
  2. Therefore, please use in the Chemical structure elucidation of the secondary metabolites Section and throughout the entire manuscript the term ”tentative identification” and do not name the compounds (harzianic acid and harzianolide), just specify the name of the fraction comprising the tentatively identified metabolites from Trichoderma sp.

Overall, the manuscript should be accepted after Minor revisions, attending the above-mentioned comments.

Author Response

(The authors gave the same response as above.)

Round 2

Reviewer 2 Report

Thank you. The Authors corrected their manuscript based on my suggestions.